# Pathogenicity and Virulence Factors of *Fusarium graminearum* Including Factors Discovered Using Next Generation Sequencing Technologies and Proteomics

**DOI:** 10.3390/microorganisms8020305

**Published:** 2020-02-22

**Authors:** Molemi E. Rauwane, Udoka V. Ogugua, Chimdi M. Kalu, Lesiba K. Ledwaba, Adugna A. Woldesemayat, Khayalethu Ntushelo

**Affiliations:** 1Department of Agriculture and Animal Health, Science Campus, University of South Africa, Corner Christiaan de Wet Road and Pioneer Avenue, Private Bag X6, Florida 1710, South Africa; rauwame@unisa.ac.za (M.E.R.); oguguatalks@gmail.com (U.V.O.); kaluchimdimang@gmail.com (C.M.K.); LedwabaL@arc.agric.za (L.K.L.); 2Agricultural Research Council-Vegetable and Ornamental Plants (ARC-VOP), Plant Breeding Division, Roodeplaat, Private Bag X293, Pretoria 0001, South Africa; 3Department of Biotechnology, Bioinformatics, Research Unit, College of Biological and Chemical Engineering, Addis Ababa Science and Technology University, Building Block 73, Room 106, Addis Ababa 1000, Ethiopia; adugnaabdi@gmail.com

**Keywords:** pathogenicity, virulence, *Fusarium graminearum*, next-generation sequencing, proteomics

## Abstract

*Fusarium graminearum* is a devasting mycotoxin-producing pathogen of grain crops. *F. graminearum* has been extensively studied to understand its pathogenicity and virulence factors. These studies gained momentum with the advent of next-generation sequencing (NGS) technologies and proteomics. NGS and proteomics have enabled the discovery of a multitude of pathogenicity and virulence factors of *F. graminearum*. This current review aimed to trace progress made in discovering *F. graminearum* pathogenicity and virulence factors in general, as well as pathogenicity and virulence factors discovered using NGS, and to some extent, using proteomics. We present more than 100 discovered pathogenicity or virulence factors and conclude that although a multitude of pathogenicity and virulence factors have already been discovered, more work needs to be done to take advantage of NGS and its companion applications of proteomics.

## 1. Introduction

Plant pathogens have developed sophisticated penetration, infection, and colonization strategies to suppress plant defense mechanisms of susceptible hosts and cause disease. The display of minor symptoms by the plant is of least concern but excessive tissue damage and crop loss result in serious economic losses with adverse implications for society, especially in poor countries. To gain insights into how these plant pathogens develop the sophisticated strategies of attack, platforms such as next-generation sequencing (NGS), and its companion applications like proteomics, are currently being exploited. Pertinent scientific inquiry of pathogenicity and virulence processes of pathogens, comprehensive and integrated investigations remain imperative and must take advantage of such high-tech platforms as NGS to generate multitudes of useful datasets. Among microbes associated with the plant are fungi which encounter plants, penetrate and colonize plant tissue to either cause disease or live within the colonized plant in a symbiotic relationship with the host. Fungi which cause diseases are called fungal pathogens and their adverse effects can range from tiny spots on the plant surface to significant symptom expression and tissue collapse. In causing diseases, fungi employ various strategies. They can penetrate the host using penetration structures and develop a large network of hyphae within the intercellular spaces. Through the hyphae, these fungi draw nutrients from the living plant cells. This mode of attack is called biotrophy. Alternatively, fungal pathogens kill the host tissue and derive nutrients from the dead tissue. This mode of attack is called necrotrophy. In the hemi-biotrophic mode of attack, the fungus successively adopts a necrotrophic lifestyle after biotrophy. The invasion of the plant by the pathogen is usually carried out through the production of factors used for plant tissue manipulation to gain physical access into the tissues as well as draw nutrients. These factors enable the plant to cause disease as well as advance the infection within the interior of the tissue, ultimately worsening the disease condition. The ability of a fungus to cause disease is termed pathogenicity, and the ability to worsen the disease is called virulence. The terms pathogenicity factors and virulence factors are loosely used to refer to any substance a pathogen uses to parasitize the plant. We will also use these concepts jointly and will not attempt to classify the various disease-causing factors as either pathogenicity or virulence factors. As a necrotrophic fungus, the wheat pathogen *Fusarium graminearum* Schwabe (teleomorph *Gibberellae zeae*) produces, among other things, cell wall-degrading enzymes (CWDEs) and toxins. Plant tissue exposed to the CWDEs and the toxins loses firmness, macerates and cell contents leak. As the cell contents provide nourishment to the fungus, it grows and advances into the inner tissues. The disease spreads and eventually, depending on other factors, the affected part dies. With its ability to produce toxins, *F. graminearum*, a homothallic (self-fertile) ascomycetous fungus, of the Order: Sordariomycetes and Family: Nectriaceae causes Fusarium head blight (FHB), one of the most economically important diseases of wheat, barley, rice and other grain crops worldwide [1,2,3,4,5]. Also known as scab, this devastating disease was first described in 1884 in England and since then, its prominence has increased worldwide with outbreaks reported in temperate and semitropical areas globally [1,6,7,8,9]. Under favorable conditions, FHB can advance from initial infection to the destruction of the entire crop within a few weeks [1]. A combination of factors, global warming, increase in wheat production under irrigation and the concomitant increase in no-till practices is the likely cause of the recent resurgence of FHB. *F. graminearum* is favored by high temperatures and irrigation splash disseminates its propagules. Wheat spikes infected by *F. graminearum* have a bleached appearance and grain infected with this fungus is shriveled with pale grey color and an occasional pinkish discoloration. Over and above these undesirable qualities, ingestion of significant amounts of mycotoxin-contaminated grain may cause vomiting, headache, and dizziness in humans. Animals may lose weight and suffer anorexia. Serious effects of ingestion of large amounts of mycotoxins include leukoencephalomalacia in horses, pulmonary edema in swine, and kidney and liver cancers in mice (mentioned in Proctor et al. [10]), and a plethora of records link mycotoxin consumption to cancer in humans. However, some of these sicknesses are linked to mycotoxins produced by other *Fusarium* species, and not *F. graminearum.* The primary mycotoxin produced by *F. graminearum* is the trichothecene deoxynivalenol (DON) (see Figure 1 for the classification of DON) and other toxins produced include zearalenone, nivalenol (NIV), 4-acetylnivalenol (4-ANIV), and DON derivatives 3- and 15-acetyldeoxynivalenol (3-ADON and 15-ADON) [2,11,12,13,14]. The serious effects of crop infection by *F. graminearum* require in-depth studies on the pathogen, primarily its pathogenicity and virulence factors which can be comprehensively studied using, among other techniques, whole-genome sequencing, transcriptomics using the convenient, reliable, and large data-generating tools of NGS and proteomics. Recently, a multitude of data has been generated for *F. graminearum* either to understand its genome organization or its gene composition as well as genes involved in plant attack. Although more work is still required, the pace at which this information is generated is too high to allow sufficient time for synthesis, organization, and communication. Meanwhile, with the dynamics of climate change, the relations between the pathogen and the plant require active and rapid utilization of the data which is generated for use to ensure plant health and ultimately good human livelihood. Given this background, it remains important to reflect on the work which has been undertaken in *F. graminearum* research and collate it as a useful resource for various *F. graminearum* workers. Furthermore, the work to understand the genome of *F. graminearum* and its pathogenicity and virulence factors requires acceleration taking advantage of NGS technologies and its supplementing proteomics applications. This is the purpose of this current review, which aimed to trace progress made in discovering pathogenicity and virulence factors, in general, as well as using NGS, and to some extent, proteomics. We present more than 100 *F. graminearum* factors which either directly perform pathogenicity and virulence functions or are indirectly linked to pathogenicity and virulence. We conclude that a multitude of pathogenicity and virulence factors have been discovered, however, more work needs to be done taking advantage of NGS and its companion applications of proteomics. This review article is organized as follows: We provide a narrative background of pathogenicity and virulence, information on *F. graminearum* and information on NGS. We then discuss in detail the various pathogenicity and virulence factors of *F. graminearum* and elaborate on the pathogenicity and virulence factors discovered using NGS (see Figure 2 for the thought process and article organization) and, to some extent, those discovered using proteomics. We collated information from various recent scientific publications to uniquely reflect the pathogenicity and virulence factors. We conclude that a multitude of pathogenicity and virulence factors have been discovered, however, more work needs to be done taking advantage of NGS and its companion applications of proteomics.

## 2. Next-Generation Sequencing, Its Relevance in Studying Plant Pathogenic Fungi and *Fusarium graminearum*

Nucleic acid sequencing has evolved from determining a few nucleotides of a nucleic acid fragment to its current form of generating large datasets of millions of reads which cover significant genomes of organisms. The generation of these large genomic and RNA datasets has been exploited by researchers to unravel various complexities of genome organization and function in many organisms. Nucleic acid sequencing is the establishment of the order of nucleotides of a given DNA or RNA fragment [15,16]. Earlier efforts to sequence nucleic acids, such as Sanger sequencing, would determine a stretch of a few nucleotides. These earlier sequencing efforts were slow and costly and the need for cheaper and faster sequencing methods grew. This demand drove the development of NGS. NGS, also termed high-throughput or massively parallel sequencing, is a genre of technologies that allow for thousands to billions of DNA fragments to be sequenced simultaneously and independently [17,18]. Although different platforms are commercially available, the NGS workflow features are generally similar, with four major steps: (i) DNA library preparation, (ii) clonal amplification, (iii) massive parallel sequencing, and (iv) data analysis [19,20]. In just the past five years, a number of instruments have been invented. The NGS platform Illumina NextSeq was invented in 2014, followed by the Ion Torrent S5/S5XL 540 in 2015. These were followed by Pacific Biosciences PacBio Sequel in 2016 [21]. These are among the instruments which have been recently invented and utilized to generate large datasets for organisms. One of the first inventions was the 454 GS FLX (454 Life Sciences, Branford, CT, USA). It utilizes an amplicon-based technology to create libraries from genetic regions selected with pairs of multiple primers [18,22,23]. On the contrary, Ion Torrent utilizes the pH variation sequencing method [18,24], whereas the modern platforms, such as Illumina MiSeq and Gene Reader System, utilize the fluorescence emission sequencing method [24,25]. NGS technologies have impressively accelerated research and are becoming employed routinely to seek solutions in different areas of research, as well as provide insights into pathogenicity and virulence of various plant pathogens [26,27,28,29,30,31,32,33,34]. Although NGS is more affordable than first-generation sequencing (considering time and money), it is still not within the reach of many laboratories in undeveloped countries [16]. However, sequencing services are offered by many sequencing companies at relatively low prices, and therefore, the need to buy the sequencers is less compelling for many. Despite the many challenges associated with NGS, such as inaccurate sequencing of homopolymer regions on certain NGS platforms which can lead to sequencing errors [16], as well as the lack of expertise in downstream processes such as data analysis, there is still success in the generation of sequences. NGS has been exploited to uncover genome organization and sequence for various plant pathogenic fungi. Plant pathogenic fungi were identified as important in some of the ambitious earlier projects in the area of characterization of fungi, such as the Fungal Genome Initiative [35,36], the Fungal Genomics Program [37,38] and its extension, the 1000 Fungal Genomes (1KFG) Project [39]. By extension, the plant pathogenic fungus *F. graminearum* has benefited tremendously and still does so from the various sequencing initiatives. The genome of *F. graminearum* has been sequenced completely and published for the benefit of *Fusarium* workers worldwide. This is in addition to the numerous gene sequences available in various databases. The complete genome and sequence resources have provided useful information on the biology, pathogenicity, and virulence of *F. graminearum*. A multitude of genes coding for pathogenicity and virulence factors of *F. graminearum* has been uncovered and the valuable information analyzed by researchers to deduce the much-needed insight into this devastating plant pathogen. We trace the important pathogenicity and virulence factors discovered and focus our attention on those which were discovered using NGS, and to some extent those which were discovered using proteomics. Within the context of this review article, pathogenicity factors and virulence factors are any substances produced by the fungus to gain access into the plant, manipulate it for its benefit, thus causing disease, as well as advance pathogenicity, thus causing the severity of the disease. Genes encoding these pathogenicity and virulence factors are themselves called pathogenicity and virulence factors. The terms pathogenicity and virulence are used jointly in this article. There does not seem to be clarity on the distinction between pathogenicity factors and virulence factors, and therefore, many publications seem to utilize these terms loosely. There is great justification to generate genome sequence information on *F. graminearum*. In order to offset the serious pathogenic problems caused by *F. graminearum* and to develop insights into the virulence and antagonistic defense mechanisms in host plants, it appears imperative to undertake the identification of the pathogenicity and virulence factors which make up the attack arsenal of *F. graminearum*. For this reason, the MIPS Fusarium graminearum Genome Database (FGDB) with an updated set with an estimated 14,000 genes and downstream analysis in a live gene validation process was established to provide a comprehensive genomic and molecular analysis [40]. Because *Fusarium* species, including *F. graminearum*, are among the most important phytopathogenic and toxigenic fungi, it is essential to understand the molecular underpinnings of their pathogenicity and virulence. Based on the comparative genomic analysis conducted on three phenotypically diverse species that include *F. graminearum*, it was revealed that among others, this particular homothallic fungal species, *F. graminearum*, has shown a relatively narrow host range that includes important crops like wheat (mentioned in Ma et al. [41]). Before discussing pathogenicity and virulence factors of *F. graminearum*, we look at pathogenicity and virulence factors of plant pathogens generally.

## 3. Pathogenicity and Virulence Factors of Plant Pathogens

Pathogenicity is the ability of an organism to cause disease, virulence, on the other hand, refers to the magnitude of the disease caused. There are three modes of attack by fungi. One mode requires ‘cooperation’ from the plant and living plant tissues. The fungus forms structures for penetration and a network of hyphae intertwined within the colonized plant tissues. Nutrients in the plants are extracted by this mass of hyphal structures formed in the intercellular spaces of the plant. This mode is called biotrophy (Figure 3). Other fungi release a barrage of degradative enzymes and toxins to lyse plant tissues and in so doing feed on the leaking cell contents and degraded tissue. This is called necrotrophy (Figure 4). Hemi-biotrophy is a hybrid mode between biotrophy and necrotrophy. Biotrophs produce proteins which circumvent the defense responses of the plant. These secreted proteins may have a counteracting effect on the plant’s defense proteins. This protein interaction in a match-match situation plays out as gene-for-gene interaction. This was found to be the case with flax rust of wheat caused by *Melampsora lini* and wheat rusts caused by the Puccinias. This gene-for-gene interaction is evidence of co-evolution between the plant and its pathogen. The mechanism of attack is less systematic in necrotrophs in which toxin production can be unspecific. Necrotrophs flood the plant with CWDEs and toxins which “chew up” the tissue. The hard physical defense structures of the plant, such as pectin, lignin, and glucans, crumble and the cells collapse. Pathogenicity and virulence-associated with plant pathogens are made possible by the secretion of pathogenicity and virulence factors which are molecules that help the pathogen colonize the plant [42,43]. For the purpose of this review, genes which are involved in the production of pathogenicity and virulence factors produced are themselves called pathogenicity and virulence factors.

Decades of research have disclosed that plant pathogens have used a notable assortment of proteins and toxins as virulence factors [44]. Several categories of pathogenicity genes, namely (i) gene signals, (ii) genes generating or detoxifying toxins, (iii) metabolic enzymes, (iv) genes involved in the generation of particular infection structures, and (v) CWDEs, can be identified (mentioned in Duyvesteijn et al. [45]). Vadlapudi and Naidu [46] have identified several fungal processes and molecules which contribute to fungal pathogenicity or virulence which have the ability to harm the plants, and these include cell-wall-degrading proteins and toxins. Reception and transduction of signals in the early phase of plant infection play a vital role in triggering development and morphogenesis mechanisms involved in the penetration of the plant host. Several plant pathogens produce toxins that can harm the tissues of plants. Toxins may be either non-specific or host selective. The link between host selective toxin (HST) secretion and pathogenicity in plant pathogens and between sensitivity to toxins and susceptibility to disease in plants provides convincing proof that HSTs can be accountable for host-selective infection and disease growth [47]. Non-specific toxins harm phylogenetically unrelated plant cells, while HSTs are both pathogenicity determinants on various hosts and their genetic varieties. *Loculoascomycetes* produce the most recognized fungal HSTs. Trichothecenes are also compounds that are toxic to plants, and they are believed to be pathogenicity and/or virulence factors. The protein region of the oomycete *Phytophthora* family of the *phytotoxin*-like scr74 gene, the C-terminal of *Phytophthora* RXLR effector paralogues and a single amino acid polymorphism in the *Phytophthora* EPIC1 effector were related to the capacity to specialize in a fresh host [48,49,50]. Stukenbrock and McDonald [51] also reported polymorphism data showing the spread of two codons in the host-specific necrotrophic effector ToxA produced by *Pyrenophora tritici-repentis* and *Stagonospora nodorum*. Various fungi are models for studying plant-directed toxins. These include *Alternaria alternata*, *Sclerotinia sclerotiorum*, *Botrytis cineria,* and *Fusarium*. Certain plant species can also be used as models to study the effect of toxins on plants as it was demonstrated with trichothecenes on the unicellular plant *Chlamydomonas reinhardtii* [52]. The production of CWDEs (e.g., cellulases, hemicellulases, xylanases, and pectinases) is one of the processes used by fungi to invade plants. CWDEs degrade the complex polymers, cellulose, hemicellulose, xylan, and pectin to gain access to the contents of the plant cell. The enzymatic action of the CWDEs crumbles the cell wall, its integrity is weakened and eventually it collapses. Coupled with toxin production, the production of CWDEs is part of the infection process of *F. graminearum* when it attacks wheat spikelets. These compounds have been reported to help the infection spread to other tissues [53]. Several pathogenicity factors identified in *Fusarium* are also part of preserved complexes or pathways, such as mitogen-activated protein kinases. However, specific genes of *Fusarium* involved in host-pathogen interactions such as HST, elicitors, or Avr genes are mainly undefined, apart from the effectors secreted in xylem. For the purposes of this review article, attention is focused on pathogenicity and virulence factors which are produced by *F. graminearum* when it infects its various hosts which are primarily grain crops.

### 3.1. Pathogenicity and Virulence Factors of Fusarium graminearum

*F. graminearum* causes FHB, a devastating disease of wheat and other crops which include maize (*Zea mays* L.) and barley (*Hordeum vulgare* L.) [1,54] (mentioned in Desjardins et al. [54]). The rate at which this fungus attacks and infects its host plant is dependent on different mechanisms involving the secretion of extracellular enzymes and mycotoxins. Wanjiru and colleagues [53] show that the pathogenicity of *F. graminearum* is dependent on the extracellular enzymes secreted. The fungus enters the host via the epidermal cell walls with the help of infection-hyphae, leading to the reduction of the host plant’s cellulose, pectin, and xylan. Upon gaining access into the plant, *F. graminearum* produces various depolymerase enzymes such as CWDEs that macerate plant tissue. Jenczmionka and Schäfer [55] showed that the production of CWDEs such as pectinases, amylases, cellulases, xylanases, proteases, and lipases by *F. graminearum* enhances its penetration and proliferation in the attacked plant. The production of CWDEs is systematic, regulated, and programmed by the pathogen during attack. To prove regulation and co-ordination, Jenczmionka and Schäfer [55] showed that the *F. graminearum* Gpmk1 MAP kinase regulates the ability of the fungus to induce extracellular endoglucanase, xylanolytic as well as proteolytic activities.

With the disruption of the cell wall, *F. graminearum* uses the polysaccharides in the plant cell wall as a source of nutrients when advancing infection into the deeper underlying tissue [56]. These CWDEs have also been reported to elicit primary immune responses in plants [57,58,59,60,61]. According to Kikot and colleagues [62], various studies have shown a link between the presence of pectic enzymes, disease symptoms, and virulence, indicating CWDEs as decisive factors in the process of phytopathogenic fungi. Plant defense requires the pathogen to engage in a more sophisticated attack mode to overcome this defense. The production of CWDEs alone does not suffice, and therefore, other pathogenicity and virulence factors must come into play. Over and above CWDEs, *F. graminearum* attacks host plants through the release of mycotoxins which include primarily the trichothecene DON, also known as vomitoxin [63]. Sella and colleagues [64] reported that *F. graminearum* secretes lipase, DON, OS-2 (a stress-activated kinase) and Gpmk1 MAP kinases as essential pathogenicity and virulent factors necessary for the development of full disease symptoms in soybean seedlings, and OS-2 is involved in overcoming the resistance possessed by the soybean phytoalexin. Voigt and colleagues [65] observed that the lipase secreted by *F. graminearum* encoded by the FGL1 gene is a virulent factor that enhances the fungal pathogenicity against wheat and maize. Through functional annotation by gene deletion, Zhang and colleagues [66] also discovered the involvement of *FgNoxR* in conidiation, germination, sexual development and pathogenicity of *F. graminearum*. In another study, Jia and colleagues [67], through gene deletion and mutant analysis, identified a putative secondary metabolite biosynthesis gene cluster *fg3_54* responsible for cell-to-cell penetration. The invasiveness of the *fg3_54-*deleted *F. graminearum* strains is restored by fusaoctaxin A (an octapeptide), leading to the conclusion that fusaoctaxin A is a virulence factor necessary for cell-to-cell invasion of wheat by *F. graminearum*. Gardiner and colleagues [68] point out that DON is associated with fungal virulence in wheat and barley and plays a crucial role in the spread of FHB. Moreover, DON elicits various defense responses in wheat which include hydrogen peroxide production, programmed cell death [69]. Trichothecenes are secondary metabolites secreted by many fungi belonging to the genera *Fusarium*, *Myrothecium*, *Stachybotrys,* and *Trichoderma*. Ueno [70] classifies trichothecenes into four types based on the unique characteristics of their chemical structures. *Fusarium* spp. secrete mostly type A and type B trichothecenes and not type C and type D [71]. These two types are characterized by the presence of a carbonyl group attached to C-8 of the sequiterpenoid backbone of trichothecenes. The presence of additional 7, 8 epoxides differentiates type C from the other types. Type D contains a macrocyclic ring that connects C-4 and C-15 of the sequiterpenoid backbone. Not a matter of discussion in the current review is the formation of toxin glycosides, also called masked mycotoxins, which are formed by cereal crops which are infected with mycotoxins. Toxin glycosides may be converted to toxins during digestion by the consumers of grain. Of great concern is the fact that toxin glycosides may escape detection by routine laboratory procedures (taken from McCormick et al. [72]. *F. graminearum* secretes type B trichothecenes DON, 15-ADON and 3-ADON or NIV. Lanoseth and Elen [73] reported the presence of DON, 3-ADON and NIV in infected oats. In *F. graminearum*-infected barley, wheat and corn, DON is the most common toxin produced by the fungus [74]. Ueno [70] indicates that the mechanism of action of trichothecenes involves binding the toxin to the ribosomes and inhibiting the synthesis of proteins.

A wide variety of proteins plays essential roles in the synthesis of trichothecenes, and most are encoded by genes within a trichothecene biosynthetic cluster (*Tri* cluster) [75,76]. The *Tri* cluster encodes 13 proteins that are responsible for the secretion of toxins in *F. graminearum*. Jenczmionka and colleagues [77] reported the role of MAP kinase Gpmk1/MAP1 in the production of *F. graminearum* conidia, which implies that the removal of the enzyme could result in a reduction in conidia production and eventually disease spread and pathogenicity. Urban and colleagues [78] observed that in their study the mutants of Δgpmk1/map1 did not cause any disease to wheat, although they were able to produce DON in the infected plant. It was further discovered that Gpmk1 was involved in the regulation of CWDEs (endo-1,4-b-glucanase and other proteolytic, xylanolytic, and lipolytic enzymes) expression [55]. Ochiai and colleagues [79] reported the activities of a protein kinase cascade in trichothecene biosynthesis. According to the authors, the protein kinases are actively involved in histidine kinase signal transduction. Alteration of the genes leads to the reduction in the expression of *Tri6*, *Tri4* and NIV production in rice cultures. Perithecia, ascospore, conidia formation and spore germination are impacted by the activities of the serine/threonine-protein kinase gene (*GzSNF1*) [80]. *GzSNF1* is also involved in the expression of *F. graminearum* genes which encode for endo-1,4-ß-xylanase 1 precursor (GzXYL1), an endo-1,4-ß-xylanase 2 precursor (GzXYL2) and an extracellular ß-xylosidase (GzXLP) which are involved in depolymerization and plant cell wall degradation. Lee and colleagues [80] also report a reduction in the virulence of Δgzsnf1 mutants on barley, proving the importance of the GzSNF1 gene in pathogenicity and virulence. The docking and fusion of secretory vesicles are required for vegetative and sexual growth in fungi. In the process of docking and fusion, v-SNARE proteins are fused with cargo proteins into vesicles. SNARE proteins are soluble N-ethylmaleimide-sensitive fusion protein (NSF) attachment protein receptors.

Hong [81] reports that the fusion of the apposing membranes which belong to the transport intermediate together with the target compartment is sped up by the interaction between the membrane-integrated t-SNARE protein and v-SNARE protein. In *F. graminearum*, syntaxin-like t-SNARE proteins are encoded by *GzSYN1* and *GzSYN2* and the deletion of these genes results in decreased virulence of the fungus on barley [82]. Syntaxin a is a multidomain protein with a globular amino terminal domain, a SNARE domain, and a carboxyl terminal transmembrane domain. In Δgzsyn1 mutants, perithecia were found to be unevenly distributed despite retaining reproductive ability with a decrease in the radial hyphal growth in culture. The Δgzsyn2 mutants are sterile females that could act like males in outcrosses with Δmat1-2 strains, resulting in the normal formation of perithecia. Radial hyphal growth of Δgzsyn2 mutants looks like the wild-type strains although the mycelia of both Δgzsyn1 and Δgzsyn2 mutants are generally thinner compared to those of wild-type strains [82].

Secreted lipase in *F. graminearum* is encoded by Fgl1 genes and the alteration of these genes in the fungus can reduce the virulence of the fungus in wheat with symptoms not spreading beyond the spikelets adjacent to inoculated spikelets [65]. Voigt and colleagues [65] further reported that in their study the maize plant infected with Δfgl1 mutants showed slight symptoms and the disease severity estimation was lower in the cobs infected with the mutant compared to the wild-type. The ability of fungi to colonize plants and initiate disease formation is dependent on the transmembrane transport proteins. These protein transporters can also exhibit virulence activities via the provision of protection from poisonous secondary metabolites secreted by host plants or even the fungus itself [83]. Many studies have indicated the roles of multiple ATP-binding cassette transporters or major facilitator superfamily (MFS) proteins in virulence activities of the fungi such as *Botrytis cinerea*, *Cochliobolus carbonum*, *Cercospora kikuchii*, Fusaria, *Magnaporthe grisea*, *Mycosphaerella graminicola* and *Penicillium digitatum* [83,84,85,86,87,88,89,90,91,92]. The same applies to *F. graminearum*, ATP-Binding Cassette Transporter Gene *FgABCC9* was found to be involved in the fungus’s pathogenicity towards wheat [93]. *F. graminearum* attacks its host plant through the secretion of mycotoxins characterized as virulence factors. These secreted virulence factors and the level of secretion depend on the presence of the relevant genes that facilitate their secretion. Presently, trichothecene mycotoxins have been identified as the major virulence factors associated with *F. graminearum* [94,95]. The trichothecene gene cluster has diverse allelic genes with some of the genes found outside of the main cluster [95]. The allelic variations in *fgTri8* are responsible for producing the DON derivatives [96]. Furthermore, allelic differences in *fgTri1* are linked to the formation of the alternate trichothecene NX-2, whereas *fgTri13* and *fgTri7* are responsible for the trichothecene NIV [97,98]. The genes associated with the trichothecene production are not the only genes discovered to be responsible for the pathogenicity and virulence of *F. graminearum*. *FGSG_04694*, a gene-encoding polyketide synthase (PKS2) responsible for mycelial growth and fungi virulence, was found in *F. graminearum* [99]. Gaffoor and colleagues [100] confirm the function of PKS2 as they observed decreased mycelial growth and virulence in a mutant PKS2 found in *F. graminearum*. PKS2 is an accessory gene with irregular patterns of conservation that is also found in *F. graminearum* responsible for secondary metabolism and virulence [101,102]. The genes within these accessory gene clusters include terpene synthase-encoding genes *FGSG_08181*, *FGSG_08182*, which encode a putative transcription factor and three putative cytochrome P450 genes *FGSG_17088*, *FGSG_08183* and *FGSG_08187* [101,102]. Harris and colleagues [103] observed the expression of these accessory genes in cereals when infected by the fungus, which implies that they are involved in the virulence of the fungus.

*ZIF1* that is conserved in filamentous ascomycetes encodes a bZIP (basic leucine zipper) transcription factor. The absence of *FgZIF1* affects the virulence and reproduction ability of *F. graminearum*. Expression of *FgZIF1* in mozif1 of *M. oryzae* mutants shows clearly that this transcription factor is functionally conserved in *F. graminearum* and *M. oryzae* [104]. Topoisomerase I encoded by *TOP1* is another enzyme involved in the pathogenicity and conidia formation of fungi. The removal of *TOP1* in *F. graminearum* was found to reduce disease expression in infected spikelets and abolition of spore formation [105]. *FGSG_10057* encodes a Zn(II)2Cys6-type transcription factor in *F. graminearum* that enhances radial growth and virulence activity of the fungus in wheat [106]. Most of these factors that enhance virulence are proteins and these proteins need to be transported to the areas in the host plant where disease symptoms are expressed. Most recently, it was found that *F. graminearum FgCWM1* encodes a cell wall mannoprotein which plays a role in pathogenicity. *FgCWM1* mutants exhibit reduced pathogenicity in wheat [107]. Another recent discovery is that of a SNARE gene *FgSec22* which was found to be required for vegetative growth, pathogenesis and DON biosynthesis in *F. graminearum* [108]. Similarly, an *F. graminearum* mitochondrial gene mitochondrial gene *FgEch1* was found to be important for conidiation, DON production, and plant infection [109]. Another recently discovered pathogenicity and virulence factor is FgPEX4, which is important for development, cell wall integrity and pathogenicity [110]. Elucidation and complete delineation of the pathogenicity and virulence factors of *F. graminearum* must still be done by increasing efforts in whole-genome sequencing of strains from various parts of the world and by conducting in planta gene expression studies to assess genes which are differently expressed during the infection of a host plant by *F. graminearum*. Nowadays these can be conveniently done using NGS. The pathogenicity and virulence factors of *F. graminearum* can be conveniently summed under CWDEs, toxins, toxin biosynthesis genes, and other pathogenicity and virulence determinants. In this section, an elaborate explanation of the different pathogenicity and virulence factors was provided. Below we provide a convenient list of these pathogenicity and virulence factors under different topics, namely, CWDEs, toxins, toxin biosynthesis genes and other pathogenicity and virulence determinants (Table 1). The pathogenicity and virulence factors are therefore summarized under their respective headings. In trying to categorize pathogenicity and virulence factors there may be some ambiguities which may lead to unclear categorizations.

### 3.2. Comparative Genomics and Molecular Basis of Pathogenicity and Virulence in Fusarium graminearum

In order to offset the serious pathogenic problems caused by *F. graminearum* and to develop insights into the virulence and antagonistic defense mechanisms in host plants, it appears imperative to undertake the identification of fungal pathogenicity and virulence factors which make up the arsenal of this fungal plant pathogen. Presently this is conveniently done using the high-throughput genome sequencing technologies which generate large datasets to reveal genome organization and the various genes present in *F. graminearum*. The information generated from genome sequencing and RNA sequencing projects can be utilized to understand the mechanisms employed by *F. graminearum* to gain entry into plant tissue as well as advance with the plant. Various commendable projects have been launched in the previous years and have advanced our understanding of pathogenicity and virulence of *F. graminearum*. Among these significant projects was the MIPS *Fusarium graminearum* Genome Database (FGDB) with an estimated 14,000 genes and downstream analysis in a live gene validation process was established to provide a comprehensive genomic and molecular analysis [40]. Because *Fusarium* species are among the most important phytopathogenic and toxigenic fungi, it is essential to understand the molecular underpinnings of their pathogenicity. In addition, the production of mycotoxins by these fungi put animals and humans who consume the crop product at risk. Given this alarming situation, aggressive research in *F. graminearum* is imperative and it must take advantage of the presence of convenient tools to generate a multitude of data to elucidate the various pathogenicity and virulence processes of *F. graminearum*. Based on the comparative genomic analysis conducted on three phenotypically diverse species that include *F. graminearum*, it was revealed that among others this particular homothallic fungal species, *F. graminearum*, has shown a relatively narrow host range which includes important cereals [41]. It is therefore important to note that this pathogen is particularly notorious on wheat, barley, rice, oats by causing head blight or ‘scab’ and on maize causing mainly stalk and ear rot disease [2]. However, genomic analysis shows that this fungus may also infect other plant species without causing disease symptoms. Further genome analysis revealed that 67 gene clusters with significant enrichment of predicted secondary metabolites and with functional enzymes were shown to be expressed among which 30% with gene overexpression were likely in virulence [102]. While the exchange of genes between the core and supernumerary genomes bestows significant opportunities for adaptation and evolution on the organism, it appears reminiscent in *F. graminearum*, to the compartmentalization of genetic material where non-conserved regions are found at various places on the four core chromosomes [111]. Studies from comparative genomics indicate that these mobile pathogenicity chromosomes exist in most *Fusarium* species with lineage-specific genomic regions [41], nevertheless, the molecular foundation of pathogenicity in *F. graminearum* was shown to be closely associated with the MAP1 gene which is also responsible for the development of perithecia in the same fungal species [78]. Furthermore, 29 *F. graminearum* genes are rapidly evolving, in planta-induced and encode secreted proteins, strongly pointing toward effector function [112], implicating genomic footprints that can be used in predicting gene sets likely to be involved in host–pathogen interactions. In association with this, as forward and reverse genetics have improved our understanding of molecular mechanisms involved in pathogenesis, it was revealed that mitogen-activated protein kinase and cyclic AMP-protein kinase A cascades both regulate virulence in *Fusarium* species and it has been postulated that cell wall integrity might be necessary for invasive growth and/or resistance to plant defense compounds [113]. These snippets which have been discovered to give clues on the pathogenicity and virulence of *F. graminearum* necessitated a deeper and comprehensive interrogation of the genome of this fungal pathogen to uncover all pathogenicity and virulence genes. NGS was conveniently used to unravel various pathogenicity and virulence factors of *F. graminearum* to the benefit of *F. graminearum* researchers worldwide.

### 3.3. Pathogenicity and Virulence Factors of Fusarium graminearum Discovered Using NGS Technologies

Understanding the molecular mechanisms involved in fungal pathogens in plants has been accelerated over the past decade. Notably, NGS has contributed immensely towards the generation of vast datasets of genomes and transcriptomes. The availability of fungal genome sequences from a majority of plant fungal pathogens has contributed to these discoveries [28,29,114,115]. Genomics, transcriptomics, proteomics, and metabolomics approaches were introduced, allowing possible identification of genes, proteins, and metabolites of fungi in various artificial cultures and during infection of plants under different experimental conditions [116,117]. Fungal pathogens cause diseases in plants, resulting in tissue damage and disease due to pathogenicity and virulence factors which assist fungal pathogen survival and persistence [118]. The pathogens affect their host by adapting to their environment and secreting/producing pathogenicity-related toxins, pectic enzymes, and hormone-like compounds. These products can have devastating effects on the quality and yield of crops in the field and can also cause postharvest diseases [119]. The mechanisms involved in fungal pathogenesis in plants are therefore being studied broadly, to protect plants against diseases of economic importance. *F. graminearum* is one of the plant pathogens that affect grain cereal crops globally, causing different diseases in different crops [80,120,121]. The pathogen produces metabolites that are toxic or non-toxic, which enables it to manipulate the plant to acquire nutrition. Inhibition of pathogenicity and virulence factors enhanced by the pathogenic fungus results in the development of diseases in plants. Pathogenicity factors involved in plant-pathogen interactions have been investigated extensively in different plants, and genes, proteins, and metabolites have been identified [81,97,122,123,124,125], including those involved in response to *F. graminearum* infections [55,104,113,120,126,127]. Pathogenicity and virulence genes that have been largely identified belong to the trichothecene biosynthesis gene cluster, as described by Proctor and colleagues [95]. For *F. graminearum*, it is largely known that pathogenicity and virulence follow a path of germ tube emergence from conidia, production of cell wall-degrading enzymes and the production of trichothecene mycotoxins [53,94,95,128].

However, complete delineation of the infection process of *F. graminearum* requires sequencing and analysis of the entire fungal genome, conveniently and preferably using NGS. From the various efforts to study pathogenicity and virulence factors of *F. graminearum* using NGS, a few studies are worth noting. The first is the study of King and colleagues [115] (sequencing was done using the Illumina HiSeq 2000 sequencing platform) which provided a complete genome sequence from a combined genome analysis from various sources which had *F. graminearum* genome sequence information. Through the modification the gene model set, the FGRRES_17235_M gene was identified to be of particular interest because it is a virulence factor, it encodes for a cysteine-rich secretory protein, allergen V5/Tpx-1-related with CAP and signal peptide domains, with a previous link with plant pathogenesis proteins of the PR-1 family [129] and had been identified in the highly virulent *F. graminearum* strain CS3005 (gene ID: FG05_09548). Another gene 15917_M was identified as an endo-1,4-beta-xylanase enzyme, which hydrolyzes (1- > 4)-beta-D-xylosidic linkages in xylans, of the cell walls. The second study by Wang and colleagues [130] identified eight genes responsible for *F. graminearum*-wheat interactions. Three of the genes had already been identified in various studies [112,131]. Their gene annotation revealed largely polymer degrading function i.e., xylanase, catalase, protease and lipase. The third study was by Cuomo and colleagues [114], who identified a variety of pathogenicity and virulence factors belonging to the gene classes cutinases, pectate lyases, pectin lyases and other genes encoding secreted proteins. The fourth study reports, among the findings, the presence of 616 potential effector genes, 126 of which are expressed in a host-specific manner. This same study by Laurent and colleagues [132] which utilized the Illumina HiSeq 2000 identified 252 variants within the genic sequences and the intergenic sequences of *Tri* genes. *Tri* genes are involved in the production of type B trichothecenes. Given the mammoth task of elucidating pathogenicity and virulence factors of *F. graminearum*, increasing efforts in whole-genome sequencing of strains from various parts of the world is necessary. These efforts must be coupled with in planta gene expression studies to assess genes which are differently expressed during the infection of a host plant by *F. graminearum*. Traditionally, common techniques for gene expression studies included northern blotting, real-time PCR and microarrays. Nowadays most studies which could be done using northern blotting, real-time PCR and microarrays can be conducted conveniently using NGS. However, these relatively old techniques paved the way for NGS.

#### 3.3.1. Notable Studies Which Paved Way for NGS

Before NGS technologies were commonplace, Northern blotting, real-time PCR, and microarrays were used to discover pathogenicity and virulence factors of fungi including *F. graminearum*. These studies paved the way for studies based on NGS. Some of the studies (reviewed in this section, none of which were performed in the past two years) utilized proteomic approaches and metabolomics. *F. graminearum* genome analysis reports that the pathogen consists of 1250 genes that encode secreted protein effectors [133]. These genes associated with fungal pathogenesis in vitro and in planta have been identified, and a large number are activated during infection. The *F. graminearum* classes of genes overlap with other plant-microbe interaction studies. These genes include the trichothecene gene cluster [95], bZIP transcription factor [104], syntaxin-like t-SNARE proteins [81], PKS [134], lipase [65], EBR1 [127], FgATG15 [135], among others. These genes play different roles in fungal pathogenesis. Among these types of genes, the *Tri* gene cluster has been characterized in the *F. graminearum* species complex, with the type B cluster being the most studied, due to their ability to cause diseases in both animals and humans [136]. In addition, proteins such as the five TRI proteins TRI1, FG00071; TRI3, FG03534; TRI4, FG03535; TRI14, FG03543; TRI101, FG07896 [137]; kinases [138] and FG00028, metallopeptidase MEP1; FG00060, KP4 killer toxin; FG00150, NADP-dependent oxidoreductase (COG2130); FG00192, peptidase S8 (pfam00082); FG00237, O-acyltransferase (pfam02458), among others [139], were found to be involved in *F. graminearum* pathogenesis [124,140,141]. The study by Dhokane et al. [124] demonstrates the interface between metabolomics and NGS. The role of the genes found to be involved in *F. graminearum* pathogenesis [124,140,141] has been identified and characterized using high throughput sequencing approaches, confirming their roles in fungal pathogenesis. NGS studies can employ either studying the genome or gene expression by means of transcriptomics. Both have been instrumental in discovering pathogenicity and virulence factors of *F. graminearum*.

#### 3.3.2. *Fusarium graminearum* Pathogenesis-Related Genes Discovered Using RNA-Seq Transcriptomics

Many technologies have been employed for the detection, identification, and quantification of mycotoxins secreted by *F. graminearum* infections in grain cereals. A few studies, including those by Pasquali and Migheli [136], report the most important fungal mycotoxins belonging to the type B trichothecenes that are produced by the *Fusarium* spp. Identification of differentially expressed genes regulated by mycotoxins using transcriptomics is one of the approaches to identify and catalog pathogenicity and virulence factors in response to *F. graminearum* in grain cereals. Using comparative transcriptomic analysis, Walkowiak and colleagues [142] identified 1500 differentially expressed genes of two *F. graminearum* strains with 3-ADON and 15-ADON trichothecene toxin chemotypes. Furthermore, a whole-genome sequencing and comparative genomics study investigated four *Fusarium* strains and reported few pathogenicity and virulence genes [99]. These included the g8968 gene, which was predicted to contain the *Tri5* domain. The *Tri5* is a terpene synthase gene that catalyzes the first step of trichothecene biosynthesis in *F. graminearum* [95]. Furthermore, *Tri8* was also identified in this study, and was reported to have exhibited a high frequency of SNPs and indels. The importance of *Tri5* in pathogenicity and virulence was also supported in non-NGS studies. Boddu and colleagues [143] report *Tri5* encoding a DON enzyme and revealed that loss-of-function *F. graminearum tri5* mutants were unable to produce DON in wheat and barley. Similar results of the importance of *Tri5* were also observed in a study by Jonkers and colleagues [144] whereby the Wor1-like Protein Fgp1 regulated pathogenicity, toxin synthesis and reproduction in *F. graminearum*. The study predicted that the loss of mycotoxin accumulation alone may be enough to explain the associated loss of pathogenicity to wheat.

Using transcriptomic analyses, differentially expressed genes (DEGs) were identified in infected spikelets and rachis wheat samples following *F. graminearum* infections [145]. From the list of the DEGs identified, a few trichothecene biosynthesis genes of the *F. graminearum Tri* cluster were mostly upregulated in the pathogen when infecting the resistant near isogeneic lines (NILs). Interestingly, another transcriptomic study was conducted between three host plants infected with *F. graminearum* strain to identify DEGs during colonization [103]. The study discovered that some genes were only expressed in a specific host, and there was also a difference in the genes’ functional categories identified in each host. In summary, the pathogenicity and virulence factors (listed in Section 3.3) of *F. graminearum* discovered using NGS technologies are provided in Table 2 below:

### 3.4. Fusarium Graminearum Pathogenesis Proteins Discovered Using Proteomics Approaches

Proteins are macromolecular machines which undertake various biochemical functions either building blocks, transporters, enzymes, and other functions. Proteins functions are coordinated, they are intertwined with other constituents of organisms like genes, RNA, and metabolites.

Proteomics is a large-scale study of sets of proteins produced by organisms. The set of total proteins produced by organisms is termed the proteome. The proteome varies across cells, and to some extent, it is defined by the underlying transcriptome. Traditionally, proteins were studied using low-throughput methods which focus on a relatively small set of proteins and provide qualitative data on structure, function, and interaction which other cell constituents. The small windows of knowledge opened by these traditional techniques denied biochemists of a broader bigger of the entire proteome of a cell. From the traditional methods of studying proteins emerged proteomics, a large-scale study of the proteome which is able to provide a snapshot of total proteins in an organism. As opposed to gel-based and antibody-based methods of studying proteins, mass spectrometry (MS) has been utilized to produce large datasets on the proteome. The basic workflow followed in proteomics is the extraction from the tissue of total proteins, followed by trypsin digestion, separation chromatography of short peptides from the digestion, and mass analysis by MS. From then onwards is the identification of proteins in the studied sample and the generation of the protein list. Similar to NGS, proteomic studies have been accelerated by the invention of various instruments which perform both the separation of the digested peptides, mass analysis, and other downstream applications. The instruments had to meet a number of qualities which include high throughput and high confidence in the identification of peptides, notably Orbitrap and time-of-flight mass analyzers [147,148,149,150]. The common trend in improving the performance of the MSs was to create hybrid systems. The hybrid systems make use of different ion analyzers or separators to enhance the capability, quality and usefulness of the results obtained. The advent of a triple quadrupole instrument enhanced MS capability over a single quadrupole. With the triple quadrupole, data on m/z values are combined with data on molecule fragmentation patterns to improve accuracy. The fragmentation pattern data is made possible by the presence of the second quadrupole which acts as a collision cell.

Fungi produce proteins for pathogenicity and virulence. Usually, some of these proteins are secreted into the intercellular spaces of plant and may either degrade the cell wall or act as effectors to perform various other pathogenicity and virulence functions. The secreted proteins are part of what is called the secretome. For comprehensive studies of pathogenicity and virulence proteins, studying the whole-organism proteome becomes necessary and it is made possible using high-throughput proteomics instruments. Yang and colleagues [138] pointed out that the invention of “omics” and bioinformatics tools has enhanced the proteome analysis of phytopathogenic fungi and their host interactions. Paper and colleagues [139] identified 120 fungal proteins of *F. graminearum* which include CWDEs from infected wheat heads through vacuum filtration. Among the identified proteins, about 56% controlled putative secretion signals. Transcriptomics data can be complemented with other omics approaches such as proteomics. The functions of proteins expressed at a given time can be identified and understood using proteomics approaches [138,151]. Although high throughput sequencing technologies have been available for over a decade, identifying differentially expressed proteins involved in fungal pathogenicity in cereals has not been widely investigated. Several proteins have been identified and characterized in *F. graminearum* and associated infections. However, they focused on the secretome and the impact of DON [137,139,152]. When the expression of *F. graminearum* proteins was investigated in response to in vitro stimulation of biosynthesis of the mycotoxin, trichothecene, 130 *F. graminearum* proteins that showed changes in expression were reported [137]. Many of the proteins identified were involved in fungal virulence. Moreover, investigation of a secretome of *F. graminearum* annotated secreted pathogenesis proteins related to the KP4 killer toxin and gEgh 16 proteins, among others, which were associated with pathogenicity [146].

Recently, a study by Lu and Edwards [153] reported about 190 small secreted cysteine-rich proteins (SSCPs) found in the genome of *F. graminearum* using genome-wide analysis. From the list of the SSCPs reported, five belonging to the cysteine-rich secretory proteins, Antigen 5 and pathogenesis-related 1 proteins were established. These SSCPs were observed to contain homologies to proteins that have established crystal structures. The authors also maintained that previous studies had not reported these SSCP associations with pathogenicity or virulence in plants. Moreover, in planta expression patterns showed upregulation of nine proteins associated with pathogenesis. These proteins contain conserved domains of Ecp-2-like panels 1 and 4, CFEM-like panel 3, Kp4-like panels 10, 14 and 9, PR-1-like panel 11, hydrophobin-like panel 12 and glycol_61 family panel 13, which are linked to fungal pathogenicity. This is in line with a study by Paper and colleagues [137], who identified 229 in vitro and 120 in planta proteins secreted by *F. graminearum* during infection of a wheat head using a high-throughput MS/MS comparative study. The study reported that 49 in planta proteins were not present in vitro, indicating that fungal lysis occurred during pathogenesis. Rampitsch and colleagues [154], on the other hand, reported 29 proteins whose relative abundance was affected in their secretome following infection by *F. graminearum* using a comparative secretome analysis. These proteins included metabolic enzymes, proteins of unknown function and pathogenesis-related proteins. Other studies involved in the identification of *F. graminearum* pathogenesis-related proteins in vitro and in planta also include the PR-3 and PR-5 proteins [155]. Various forms of proteomics are significant in studying plant-pathogen relations and other factors which include elicitors. An example of this is phosphoproteomics which has to some extent been studied in necrotrophic pathogens like *B. cinerea* [156,157,158], *Septoria tritici* [159]. These studies need to be extended to *F. graminearum* to deepen the understanding of its pathogenicity and virulence.

## 4. Summary and Conclusions

The infection mechanism of the plant pathogenic fungus *F. graminearum* is complex and intricate. It involves the production of a germ tube from conidia, the production of CWDEs, and eventually, the production of mycotoxins, chiefly type B trichothecene mycotoxins, DON, and NIV. Within the context of this review article, we regarded the CWDEs and the mycotoxins as pathogenicity and virulence factors which enable *F. graminearum* to gain entry into the plant and advance within the interior of the infected tissue. Our focus was on pathogenicity and virulence factors so far discovered, and we also devoted attention to those discovered using NGS and, to a limited extent, proteomics. We conclude that a multitude of pathogenicity and virulence factors have been discovered, however, more work needs to be done taking advantage of NGS and its companion applications of proteomics. Discovery of more pathogenicity and virulence and factors may facilitate the newer methods of control of *F. graminearum* infection of wheat and DON accumulation, for instance, as it has been shown by Machado et al. [160] using RNAi. Progress on the use of RNAi may depend greatly on the discovery of more pathogenicity and virulence factors of *F. graminearum*.

## Figures and Tables

**Figure 1 microorganisms-08-00305-f001:**
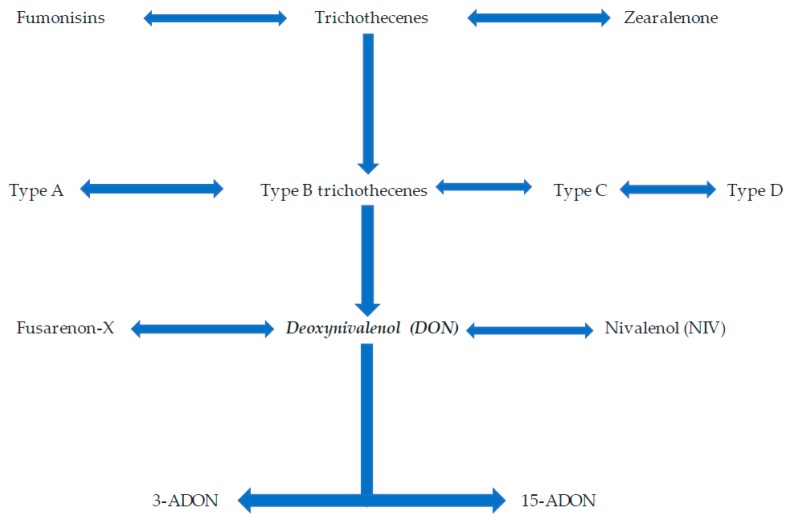
The classification of the major toxin produced by *Fusarium graminearum*, DON. The major toxins produced by *Fusarium* species are fumonisins, trichothecenes, and zearalenone. DON is a type B trichothecene which has derivatives 3-ADON and 15-ADON.

**Figure 2 microorganisms-08-00305-f002:**
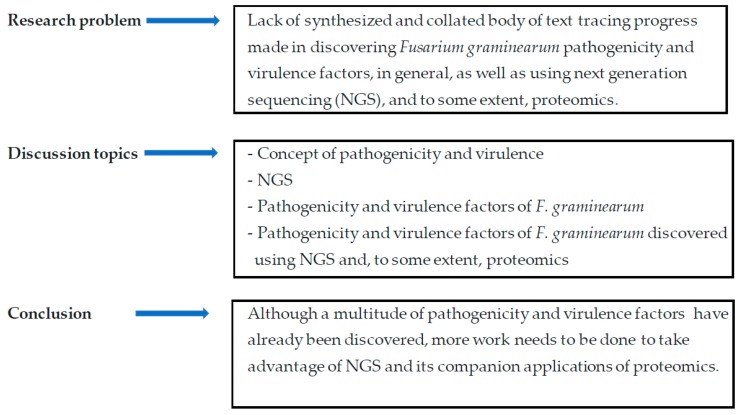
An illustration of concept development and the organization of the article.

**Figure 3 microorganisms-08-00305-f003:**
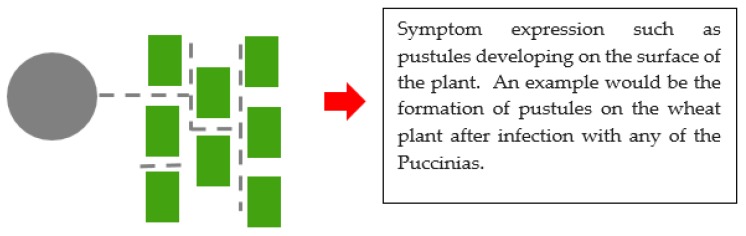
An illustration of biotrophy between a plant pathogenic fungus and a plant. The grey circle represents a fungal spore. Dotted lines represent the germ tube which has developed to form a network of hyphae, and the green rectangles represent plant cells. Finally, symptoms are expressed.

**Figure 4 microorganisms-08-00305-f004:**
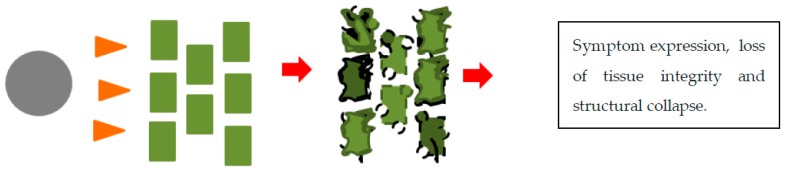
An illustration of necrotrophy between a plant pathogenic fungus and a plant. The grey circle represents a fungal spore, the orange arrows represent secreted pathogenicity and virulence factors. The green rectangles represent plant cells, the plant cells become deformed, ripped, and cell contents leak. Finally, symptoms are expressed.

**Table 1 microorganisms-08-00305-t001:** List of pathogenicity and virulence factors of *Fusarium graminearum*.

Cell Wall Degrading Enzymes			
	Category/Type/Classification	Function	At Least One Reference Where Mentioned
Depolymerase enzyme	Degradative enzyme	Catalyses depolymerization reactions	[53]
Pectinase	Degradative enzyme	Breaks down pectin	[55]
Cellulase	Degradative enzyme	Breaks down cellulose	[55]
Extracellular endoglucanase	Degradative enzyme	Breaks down glucan	[55]
Endo-1,4-b-glucanase	Degradative enzyme	Breaks down glucan	[55]
Proteolytic enzyme	Degradative enzyme	Breaks down proteins	[55]
Xylanolytic enzyme	Degradative enzyme	Breaks down xylan	[55]
Lipolytic enzyme	Degradative enzyme	Breaks down lipids	[55]
**Toxins**			
Trichothecene NX-2	Type A trichothecene (toxin)	Toxicity	[97]
Deoxynivalenol (DON)	Type B trichothecene (toxin)	Toxicity	[2]
3-acetyldeoxynivalenol (3-ADON)	Type B trichothecene (toxin)	Toxicity	[11]
15-ADON	Type B trichothecene (toxin)	Toxicity	[11]
Nivalenol (NIV)	Type B trichothecene (toxin)	Toxicity	[11]
Fusaoctaxin A	Toxin	Responsible for cell-to-cell invasion of wheat by *F. graminearum*	[67]
**Toxin biosynthesis genes**			
*Tri* cluster genomic region	Trichothecene biosynthesis genes	Involved in the synthesis of trichothecene mycotoxins	[75]
**Other pathogenicity and virulence determinants**			
Gpmk1 MAP kinase	MAP kinase	Involved in mating, conidiation, and pathogenicity	[77]
OS-2	Stress activated kinase	Involved in conferring resistance to a soybean phytoalexin	[64]
*FGL1*	Lipase gene	Enhances the fungal pathogenicity against wheat and maize	[65]
*FgNoxR*	A regulatory subunit of NADPH oxidase	Involved in conidiation, sexual development and pathogenicity of *F. graminearum*	[66]
*fg3_54*	Putative secondary metabolite biosynthesis gene cluster	Responsible for cell-to-cell invasiveness	[67]
Protein kinase	Kinase cascade in trichothecene biosynthesis	Involved in trichothecene production	[79]
Histidine kinase	Kinase	Involved in trichothecene production	[79]
GzSNF1	Serine/threonine-protein kinase	Responsible sexual, asexual development, and virulence	[80]
*GzXYL1*	Endo-1,4-ß-xylanase 1 precursor gene	Involved in plant cell wall degradation	[80]
*GzXYL2*	Endo-1,4-ß-xylanase 2 precursor gene	Involved in plant cell wall degradation	[80]
*GzXLP*	Extracellular ß-xylosidase gene	Involved in depolymerization and plant cell wall degradation	[80]
v-SNARE protein	Vesicle related proteins SNARE	Interacts with t-SNARE to catalyze the fusion of the apposing membranes of the transport intermediate and the target compartment)	[82]
t-SNARE protein	Target membrane-related SNARE	Interacts with v-SNARE to catalyze the fusion of the apposing membranes of the transport intermediate and the target compartment	[82]
Syntaxin-like t-SNARE protein	Syntaxin-like membrane-integrated protein	Required for vegetative growth, sexual reproduction, and virulence in *Gibberella zeae.* Proteins are also encoded by *GzSYN1* and *GzSYN2* in *F. graminearum* that enhanced virulence of the fungus on barley	[82]
*GzSYN1*	Syntaxin-like SNARE gene	Enhances perithecia and radial hyphal growth	[82]
*GzSYN2*	Syntaxin-like SNARE gene	Enhances perithecia and radial hyphal growth	[82]
Multiple ATP-binding cassette transporter	Major facilitator superfamily of membrane transporter	Involved in virulence	[83]
*FgABCC9*	ATP-binding cassette transporter gene	involved in the fungal pathogenicity towards wheat	[93]
*FGSG_04694*	Gene-encoding polyketide synthase PKS2	Responsible for mycelial growth and fungi virulence	[99]
*PKS2*	Polyketide synthase gene	Responsible for secondary metabolism and virulence	[101]
*FGSG_08181*	Terpene synthase-encoding gene	Involved in the virulence of the fungus	[99]
*FGSG_08182*	Terpene synthase-encoding gene	Involved in the virulence of the fungus	[103]
*FGSG_17088*	Putative cytochrome P450 gene	Responsible for the expression of disease in fungus-infected cereals	[99]
*FGSG_08183*	Putative cytochrome P450 gene	Responsible for the expression of disease in fungus-infected cereals	[99]
*FGSG_08187*	Putative cytochrome P450 gene	Responsible for the expression of disease in fungus-infected cereals	[99]
*ZIF1*	Encodes bZIP transcription factor	Involved in virulence and reproduction ability of *F. graminearum*	[104]
bZIP transcription factor	Transcription factor	Enhances the virulence of *F. graminearum* in infected plants	[104]
*TOP1* I	Enzyme	Involved in sporulation and pathogenicity	[105]
*FGSG_10057*	Conserved hypothetical protein	Involved in growth and virulence	[106]
Zn(II)2Cys6-type transcription factor	Transcription factor	Regulates fungal reproduction and pathogenicity	[106]

**Table 2 microorganisms-08-00305-t002:** List of pathogenicity and virulence factors of *Fusarium graminearum*, including genes discovered using comparative genomics methods.

Cell Wall Degrading Enzymes			
	Category/Type/Classification	Function	At Least One Reference Where Mentioned
15917_M	Endo-1,4-beta-xylanase enzyme	Hydrolyses (1- > 4)-beta-D-xylosidic linkages in xylans of the cell walls	[115]
Xylanase	Degradative enzyme	Bring about the disintegration of xylan and cell wall penetration	[55]
Protease	Degradative enzyme	Responsible for the breakdown of protein	[55]
Lipase	Degradative enzyme	Responsible for the breakdown of lipids	[55]
Cutinases	Degradative enzyme	Plays polymer degrading function	[114]
Pectate lyases	Degradative enzyme	Plays polymer degrading function	[114]
Pectin lyases	Degradative enzyme	Plays polymer degrading function	[114]
β-amylase protein	Degradative enzyme	Involved in *F. graminearum* pathogenesis	[138]
Metallopeptidase	Degradative enzyme	Involved in *F. graminearum* pathogenesis	[139]
Peptidase	Degradative enzyme	Involved in *F. graminearum* pathogenesis	[139]
**Toxins**			
Type B trichothecenes	Trichothecene mycotoxin	Toxicity	[2]
KP4 killer toxin	Toxic polypeptide	Toxicity	[146]
**Genes for toxin biosynthesis**			
TRI1 gene	*Tri* cluster gene	Involved in *F. graminearum* pathogenesis	[137]
FGRRES_17235_M	Virulence-related gene	Encodes cysteine-rich secretory protein, allergen V5/Tpx-1-related with CAP and PR-1 family	[115]
15917_M	Endo-1,4-beta-xylanase enzyme	Hydrolyses (1- > 4)-beta-D-xylosidic linkages in xylans of the cell walls	[115]
*g8968* gene (predicted to contain the *Tri5* domain)	Pathogenicity and virulence gene	Predicted to contain the *Tri5* domain	[99]
*Tri5*	*Tri* cluster gene	Involved in trichothecene biosynthesis	[95]
*Tri8*	*Tri* cluster gene	Involved *Fusarium* trichothecene phytotoxicity	[96]
**Pathogenicity and virulence proteins**			
TRI3	Trichothecene biosynthesis protein	Involved in *F. graminearum* pathogenesis	[137]
TRI4	Trichothecene biosynthesis protein	Involved in *F. graminearum* pathogenesis	[137]
TRI101	Trichothecene biosynthesis protein	Involved in *F. graminearum* pathogenesis	[137]
**Other pathogenicity and virulence determinants**			
Hormone-like compounds	Compounds with hormone-like properties	Enhances the adaptation of the fungi to the host plant environment	[119]
PR-1 family proteins	Pathogenicity related protein	Involved in pathogenicity	[129]
Basic leucine zipper (bZIP) transcription factor	Transcription factor	Enhances the virulence and reproduction ability of *F. graminearum* in infected plants	[104]
Syntaxin-like t-SNARE proteins	Syntaxin-like membrane-integrated proteins	Required for vegetative growth, sexual reproduction, and virulence in *G. zeae*. Proteins are also encoded by *GzSYN1* and *GzSYN2* in *F. graminearum* that enhanced virulence of the fungus on barley	[82]
Polyketide synthase (PKS) gene	Polyketide synthase gene	Responsible for secondary metabolism and virulence	[101]
Enhanced branching 1 (EBR1)	Zn(2)Cys(6) transcription factor	Involved in *F. graminearum* pathogenesis	[127]
NADP-dependent oxidoreductase	Oxidoreductase enzyme	Involved in *F. graminearum* pathogenesis	[139]
O-acyltransferase	Transferase enzyme	Involved in *F. graminearum* pathogenesis	[139]
Wor1-like Protein Fgp1	Regulatory protein	Regulates pathogenicity, toxin synthesis, and reproduction in *F. graminearum*	[144]

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
