# Peer review of "Pathogenicity and Virulence Factors of Fusarium graminearum Including Factors Discovered Using Next Generation Sequencing Technologies and Proteomics"

_microorganisms, 2020, doi:10.3390/microorganisms8020305_

Round 1

Reviewer 1 Report

F. graminearum is an important plant pathogen. In the manuscript authors described pathogenicity and virulence factors of the pathogen, including these discovered using NGS and proteomisc. The work is relevant though not well organized and sometimes difficult to understand. To make it more clear some corrections are needed.

Authors describe NGS in a separate paragraph. How about proteomics?Such paragraph describing its importance for pathogenesis studies should be included.

Paragraph 3 and 4 should be combined as in both of them virulence genes are described. However, I recommend dividing it into subsections: toxins, CWDES, other virulence determinants, regulation of virulence factors. Also a table summarizing all virulence factors, their role, brief description and reference is required. Some proteins should be described in more detail e.g. SNARE, ZIF1 ( here put the full name of b-ZIP in the bracket).

An update of virulence factors is also required as recent work indicate the role of other determinants: FgCWM1, FgSec22, FgEch1, FgPEX, RNAi machinery( DICER-like and Argonaute).

Author Response

Comments:

F. graminearum is an important plant pathogen. In the manuscript authors described pathogenicity and virulence factors of the pathogen, including these discovered using NGS and proteomisc. The work is relevant though not well organized and sometimes difficult to understand. To make it more clear some corrections are needed.

Authors describe NGS in a separate paragraph. How about proteomics?Such paragraph describing its importance for pathogenesis studies should be included.

Paragraph 3 and 4 should be combined as in both of them virulence genes are described. However, I recommend dividing it into subsections: toxins, CWDES, other virulence determinants, regulation of virulence factors. Also a table summarizing all virulence factors, their role, brief description and reference is required. Some proteins should be described in more detail e.g. SNARE, ZIF1 ( here put the full name of b-ZIP in the bracket).

An update of virulence factors is also required as recent work indicate the role of other determinants: FgCWM1, FgSec22, FgEch1, FgPEX, RNAi machinery( DICER-like and Argonaute).

Response:

Reviewer 1 made a comment that the work is not well organised.  We have improved the organisation of the article by summarizing/providing lists of the discussed pathogenicity and virulence factors under the main subsections.  This helps the reader to recap before moving to subsequent sections.  We have also reorganized our numbering and merged the sections which required merging.  A section on proteomics was added as suggested and it is highlighted green in the manuscript.  We have combined the paragraphs the reviewer suggested they be combined and we have provided convenient lists under the headings CWDEs, toxins and other virulence determinants.  Pathogenicity and virulence factors such SNARE, ZIF1 and b-ZIP whose abbreviations required writing out in full, and with brief descriptions, were clearly explained.  An update was provided to indicate the role of the determinants FgCWM1, FgSec22, FgEch1, FgPEX and RNAi machinery.  Articles which contain this information were incorporated into the manuscript and the extracted information blends well with the rest of the text.  All incorporated articles are highlighted in green in the improved article version.

Reviewer 2 Report

Manuscript reviews the current state of the art on F. graminearum pathogenicity and virulence factors discovered by traditional methods, massive sequencing analisys (DNAseq, RNAseq) and proteomic approaches.
Manuscript is well written, updated and drive the reader from the beginning knowledge on F. graminearum and its pathogenic modus operandi to more than 80 pathogenicity factors discovered during years. Through this manuscript, authors provide a critical point of view on NGS and encourage to use these tecniques as the best metods, at the present , to produce large amount of data to work on.

I didn't find particular issues to manage, however, I suggest to improve the paper with two informations:
- in chapter 3.1 make mention of the role of DON as oxidative stress inducer (Desmond et al. 2008 The Fusarium mycotoxin deoxynivalenol elicits hydrogen peroxide production, programmed cell death and defence responses in wheat)
- to include T2 and HT2 in text and Figure 1 among mycotoxins to take into account . Although the spread of these toxins is limited if compared to DON and NIV, and their role is not defined during colonization process, the phytotoxic activity (Alexander et al. 2008 Phytotoxicity of selected trichothecenes using Chlamydomonas reinhardtii as a model system; Mc Cormick et al 2015 Anomericity of T-2 Toxin-glucoside: Masked Mycotoxin in Cereal Crops) can provide some clues about their role during infection.

Moreover, please correct "Jia and bcollegues" at line 267

Based on my opinion and hoping my suggestions will be appreciated and implemented, I consider the manuscript acceptable for publication after minor revision.

Best regards

Author Response

Dear Editor

The issues are dealt with. 

Reviewer 2 suggested that we mention that DON is an oxidative stress inducer (Desmond et al., 2008).  We have incorporated that information, see green highlight, and the referencing has been updated.  The reviewer also suggested that Figure 1 reflects T-2 and HT2.  We welcome this as a valuable suggestion however in an article which aims to present a full chart of all the types of toxins produced by Fusarium species.  For the purposes of the present review we cannot include T-2 and HT2 on Figure 1 because T-2 and HT2 are members of type A trichothecenes which were not the focus of this review.  Figure 1 was meant to just position DON on the bigger chart of mycotoxins.  This is because DON is the major toxin produced by F. graminearum.  The reviewer pointed us to Alexander et al., 2008 to emphasize a point about T-2 and HT2, we have incorporated Alexander et al., 2008 to demonstrate the use of fungi as models to study trichothecene-induced phytotoxicity.   Mc Cormick et al 2015 (also pointed by the reviewer) was also incorporated since we wanted to highlight the complexity of toxicoses related to mycotoxins.  Jia and bcollegues was corrected in the manuscript.

Regards,

Khayalethu

Reviewer 3 Report

This review is a compilation of the updated publications of the phytopathogenic fungus Fusarium graminearum. It includes the most important contributions related to the pathogenicity and virulence factors of this phytopathogen and others whose attack system may be similar.

It is an important review, taking into account not only the losses caused by this phytopathogen but also the mycotosines it produces and its relevance in the Agrifood sector.

The review includes not only the classical techniques but the most modern "omics" techniques such as the next generation sequencing.

Since the review cites the phytopathogen Botrytis cinerea, the work would be enriched with any of these biliographic citations:

Eva Liñeiro, Cristina Chiva, Jesús M. Cantoral, Eduard Sabidó, Francisco Javier Fernández-hacerO “Phosphoproteome profile of B. cinerea under different pathogenicity stages by plant based elicitors”. Journal of Proteomics 139: 84-94. 2016

Eva Liñeiro, Cristina Chiva, Jesús M. Cantoral, Eduard Sabidó, Francisco Javier Fernández-Acero. Modifications of fungal membrane proteins profile under pathogenicity induction: a proteomic analysis of Botrytis cinerea membranome Proteomics 16 (17): 2363-2376. 2016. doi:10.1002/pmic.201500496.

Eva Liñeiro, Cristina Chiva, Jesús M. Cantoral, Eduard Sabidó, Francisco Javier Fernández-AcerO. Dataset of the Botrytis cinerea phosphoproteome induced by different plant-based elicitors. Journal of Proteomics 7. 1447-1450. 2016. http://dx.doi.org/10.1016/j.jprot.2016.03.019

Author Response

Dear Editor

The issues are sorted. 

Reviewer 3 listed a number articles which could enhance the article.  Truly, Botrytis has the same mode of attack as F. graminearum and therefore the Botrytis articles are very relevant hence we were able to incorporate all the three suggested articles together with a Septoria article. 

Regards,

Khayalethu

Round 2

Reviewer 1 Report

The autorskim applied all the reviewer’s suggestions therefore I recommend publishing the paper in the present firm.

Author Response

All issues were incorporated.